# Selenium Status: Its Interactions with Dietary Mercury Exposure and Implications in Human Health

**DOI:** 10.3390/nu14245308

**Published:** 2022-12-14

**Authors:** Ujang Tinggi, Anthony V. Perkins

**Affiliations:** 1Queensland Health Forensic and Scientific Services, 39 Kessels Road, Coopers Plains, QLD 4108, Australia; 2School of Health, University of the Sunshine Coast, Sippy Downs, QLD 4456, Australia

**Keywords:** selenium, mercury, dietary intakes, blood levels, trace elements

## Abstract

Selenium is an essential trace element in humans and animals and its role in selenoprotein and enzyme antioxidant activity is well documented. Food is the principal source of selenium, and it is important that selenium status in the body is adequately maintained for physiological functions. There has been increasing attention on the role of selenium in mitigating the toxic effects of mercury exposure from dietary intake in humans. In contrast, mercury is a neurotoxin, and its continuous exposure can cause adverse health effects in humans. The interactions of selenium and mercury are multi-factorial and involve complex binding mechanisms between these elements at a molecular level. Further insights and understanding in this area may help to evaluate the health implications of dietary mercury exposure and selenium status. This review aims to summarise current information on the interplay of the interactions between selenium and mercury in the body and the protective effect of selenium on at-risk groups in a population who may experience long-term mercury exposure.

## 1. Introduction

Ever-increasing global industrialisation has significantly contributed to the release of chemical contaminants, including selenium (Se) and mercury (Hg), into the environment. The release and disposal of industrial wastes and coal-burning plants have significantly contributed to the global pollution of Hg and Se, and this will eventually bioaccumulate in the food chain and subsequently into humans from dietary exposure [1,2]. Of particular concern is the dietary exposure from high marine fish consumption, which is frequently being raised and debated in the media regarding the risk of high Hg exposure, the benefits of the nutritional importance of omega-3 in fish, and the presence of essential trace elements such as Se [3,4].

Se is an essential trace element that plays an important role in antioxidant selenoenzymes, and its low status in humans has been reported to be linked with increased risk of cancer, heart diseases and impaired immunity [5]. High protein foods such seafood and meat products are major sources of Se, and some marine fish such as sharks and swordfish can contain high levels of Se and Hg as predatory fish are at the top of the marine food web [6,7]. Different food types can contain different forms of Se compounds, but predominantly as selenomethionine, which is also more bioavailable than the inorganic forms of selenites and selenates [5].

In contrast, Hg is a highly toxic element and its adverse health effects in humans are well documented. The Hg poisoning incident in humans, known as Minamata disease in Japan, is well documented after consumption of contaminated fish, and the consumption of flour accidentally contaminated with organic Hg in Iraq [8,9]. Concerns about Hg poisoning has led to the establishment of the Minamata Convention to provide information on reducing Hg exposure and to safeguard the wellbeing of human populations around the world [10,11]. The levels of Hg in most foods are generally low, and it is predominantly present in marine fish and seafood in the form of methylmercury, which is lipid soluble and highly toxic [12]. There have been many studies on the protective effects of Se and its related chemical species against Hg accumulation in animal body tissues, including fish, but the results are often controversial, and the possible mechanism for the modification of Se species on Hg accumulation is extremely complex and not well understood [13]. However, it has been suggested that the reasons behind the protective effects of Se species on Hg toxicity may include the distribution of Hg into various tissues, the competition for binding sites, and the formation of Hg–Se complexes or prevention of oxidative stress [14,15,16]. It has been reported that Se can form complexes with Hg in biological tissue such as tiemannite, Se–Hg–S species or Se–Hg precipitates [17,18].

There has been increased interest in investigating the interaction between Se and Hg in fish, and in particular the effects of Se concentrations and their chemical forms on Hg bioavailability and the use of molar ratios of Se and Hg concentrations for assessment of Hg toxicity [13,19]. However, there is also a concern that high Hg exposure from diets, especially marine fish diet, could also cause a reduction in the activity of selenoenzymes as a result of strong inhibition by methylmercury, and Se depletion due to the formation of insoluble Hg selenides [20]. The complexity of Se and Hg interactions has attracted interest over recent years, especially in developing sensitive analytical methods for determining Se and Hg speciation, which are important to further understanding their interactions and bioavailability in biological systems [17,21,22]. There have been few studies on the bioavailability of Se and Hg and their chemical forms in humans from simultaneous exposure of these elements from food [17,23]. This paper aims to review the current knowledge on Se status and its abilities to interact with Hg exposure through dietary intake and to mitigate Hg toxicity that could have implications for human health, particularly for the high-risk groups in a population.

Since this paper aims to provide a brief overview of the interrelationship between Se status and Hg exposure in humans, a broad literature search was performed with PubMed and ScienceDirect on the topic. The literature search was conducted in August and September 2022 for the latest information on Se–Hg interaction relating to human health. For the PubMed literature search, this included the keywords or combinations of them related to “Se status”, “Se dietary intakes”, “Se blood”, “Hg exposure”, “Hg dietary intakes”, “Hg blood”, “Se biomarkers”, “Hg biomarkers”, “healthy human population”, “Se and Hg interactions”, and “gene expression”. The search for specific “selenium status AND mercury status AND humans” produced 149 results and the selection of the articles was based on their relevance to human health. The specific search for “Se AND Hg AND interaction AND humans” produced 169 articles, and the selections of these articles was based on Se–Hg interaction relating to human exposure and health.

For the ScienceDirect search, this included the keywords or combinations of them related to “Se chemistry”, “Hg chemistry”, “Se speciation”, “Hg speciation”, “food analysis”, “diet”, “fish”, “seafood”, “Se levels, Hg levels”, “blood”, “urine”, “hair”, and “biological tissue”. The specific search for “Se AND Hg AND speciation AND food AND humans”, excluding other animals, came with 166 results, and the selection of the articles was based on relevance to food intakes and exposure in humans. These searches also included names of authors and the area of their expertise and specialisation.

## 2. Chemical Characteristics and Properties of Se and Hg

Se belongs to group 16 and in position VIA between sulphur and tellurium in the periodic table. It is classified as metalloid (semimetal) with the atomic mass of 78.96 and it has six naturally occurring isotopes (Se-74, Se-82, Se-76, Se-77, Se-78, Se-80) [24]. In the ground state, Se has the electronic configuration of ([Ar] 4s^2^3d^10^4p^4^), which utilises six valence electrons and two unpaired electrons to form covalent bonds. Se can form into five allotropic structures consisting of two amorphous and three crystalline states [24,25]. Se is present in the oxidation states of Se (0), Se (-II), Se (IV) and Se (VI), which are important for forming various Se compounds in biological systems, including selenocysteine, which forms the active sites of selenoproteins with predominantly antioxidative activities [24].

Hg, with the atomic mass of 200.59, belongs to group 12 of the periodic table and is often referred to as liquid silver [26]. It has seven stable isotopes (Hg-196, Hg-198, Hg-199, Hg-200, Hg-201, Hg-202, Hg-204) [26]. Hg has the electronic configuration of [Xe]4f^14^5d^10^6s^2^, and the 6s^2^ electronic pair determines its low chemical reactivity. The presence of the occupied d-sublevel orbital allows Hg to form covalent bonds with halides, thiol (-SH) and selenol (-SeH) groups. Hg can also form various organic and inorganic Hg compounds, as in di-, tri- and tetranuclear clusters (Hg_2_^2+^, Hg_3_^2+^, Hg_4_^2+^) [27].

## 3. Sources of Se and Hg in the Environment

The release of Se and Hg into the environment can be the result of anthropogenic and natural causes. The emission from volcanic eruptions, withering of rocks and ground water movement and degradation of vegetation are the natural causes of Se and Hg release into the environment and atmosphere [24,28]. In the aquatic environment, Se is readily present as soluble selenites (Se (IV)) or selenates (Se (VI)), and these are taken up by plants and metabolised into different forms of Se compounds, including volatile selenide and methylated Se compounds, which will eventually be released into the atmosphere [24]. Human activities can also contribute significantly to the anthropogenic sources of Se, where it is widely used in industry, including the production of electronic components, agricultural products (e.g., fertilizers, pesticides, livestock feed) and pharmaceutical and medical appliances [29].

Hg is widely used in industry for manufacturing electronic devices, paint, fungicides and waste incarnation, and subsequently these are released into the environment. The emissions from coal-burning fuel and mining activities are also major anthropogenic sources of Hg released into the atmosphere [30]. Hg is also naturally released as elemental Hg from volcanic ash, withering of rocks, forest fires, soil and vegetation volatilisation. In the environment, Hg predominantly exists in three valance states as elemental Hg (Hg (0), monovalent or mercurous Hg (Hg (I)), and divalent or mercuric Hg (Hg (II)), which can form into other inorganic and organic complexes [26,31]. The presence of Hg (I) is unstable and it readily undergoes redox processes to form elemental Hg and Hg (II), and becomes further metabolised by microorganisms to form methylated Hg as methyl and dimethyl Hg [32]. The methyl Hg is the most toxic form of Hg compound, and has complex mechanistic actions including its ability to form a low-molecular weight with thiol (-SH) that can readily transport into cells and disrupt protein functions in biological tissues [33,34].

## 4. Se Status and Hg Exposure

Dietary intake is the principal source of Se and the levels in foods can vary significantly depending on the Se levels in soil, which are taken up by plants and subsequently transfer to animals and humans via the food chain. The levels of Se in typical food groups are presented in Table 1.

There have been significant variations in reported Se intakes in human populations from different countries. Low Se status has been reported in countries such as New Zealand and Finland due to low levels in soils and crops. Importation of Australian wheat containing high Se content has improved the Se status in the New Zealand population [39]. In Finland, the Se-fortified fertilisers have been introduced and used in crops to improve the Se status in the population [40]. Low Se status has also occurred in some parts of the Congo, central Serbia and parts of China [41]. Extreme cases of Se deficiency in humans have caused Keshan disease in China, which is associated with cardiomyopathy, and Kashin–Beck disease, associated with degenerative osteoarthropathy [42].

In another extreme example in China, high levels of Se intake has been observed in a population from consuming food crops growing in high-Se soil that could pose a risk of Se poisoning [43]. In the USA, high-Se status has been reported in the South Dakota population where food crops are produced in an area of high-Se soil [44]. In diets, Se is taken up predominantly as both organic (selenomethionine, selenocysteine) and inorganic forms (selenites, selenates) [41]. The increased use of vitamin and mineral supplements, including Se supplements, can also contribute to increased intake of and exposure to Se. The gap between Se essentiality and toxicity is relatively narrow, and over exposure of Se is a cause of concern due to toxic effects. The risk of a high intake of selenomethionine (as L-selenomethionine), which is a commonly used Se compound in food supplements, has been comprehensively reviewed and assessed, and it has been suggested that the addition of selenomethionine in food supplements would not be a safety concern in adults at levels up to 250 µg/day, which is equivalent to 100 µg Se/day [45]. This value of 100 µg Se/day from the intake of food supplements is considerably below the Se Upper Tolerable Limit Level at 300 µg Se/day for Europe, which is also below the Upper Tolerable Intake Level for Se in the USA, at 400 µg Se/day [45,46]. The bioavailability of Se intake can be influenced by its chemical forms, and the interactions with trace elements (zinc, cadmium) and other nutrients such as methionine-containing proteins which affect the absorption in the body [47,48]. Generally, selenomethionine and selenocysteine, which are the common forms in food, are more bioavailable than the inorganic forms (selenate or selenite) [49]. In the body, the ingested compounds from diet will undergo methylation and dimethylation, and subsequently be exhaled in the breath as dimethyl selenide and excreted in urine as selenosugars and trimethylselenide [50]. The typical levels of Se dietary daily intakes of healthy populations from selected countries are shown in Table 2.

The variation in Se dietary intakes between countries has recently been reviewed [59]. The association between Se intake and adverse health effects can be presented by a U-shaped response that could indicate various clinical conditions [60]. Concern about adverse health effects has led many countries to provide recommended guidelines for adequate levels of Se daily intakes. For instance, in Australia the recommended Se daily intake for men is 70 µg/day and 60 µg/day for women, with an increase to 65 µg/day for pregnancy, and the upper limit (UL) intake at 400 µg/day [61].

With respect to mercury, the levels of Hg in foods are generally low, except for marine fish and shellfish. However, crops that grow in Hg contaminated soil can accumulate Hg from plant uptake, which can contribute to Hg exposure in human population. The levels of Hg in selected foods from selected countries are shown in Table 3.

Rice crops grown near mining areas have been shown to contain considerable amounts of Hg and could pose health risks from rice consumption; rice could contribute to Hg daily intakes in the range from 0.042 to 0.35 μg/kg bw/day, which is equivalent to 2.52 to 21.0 μg/day for an adult at 60 kg body weight (bw) [65,66]. Of greater concern is the consumption of marine fish and shellfish, which have been known to accumulate Hg in the form methyl Hg, which is a neurotoxin. The levels of methyl Hg in fish can vary significantly, and the highest levels are found in larger and older predatory fish at the top of the food web (e.g., sharks, swordfish, tuna) [67,68]. The consumption of fish and shellfish can contribute significantly to dietary exposure to methyl Hg in a population. In Europe it has been established that the tolerable weekly intake (TWI) for methyl Hg is set at 1.3 µg/kg body weight, which is equivalent to 13 µg/day as Hg for an adult weighing 70 kg [69]. Table 4 shows a typical daily Hg intake of a healthy population from selected countries.

## 5. Blood Se and Hg Levels

The trace element levels in blood, including Se and Hg, have been widely used in assessing the trace element status in a population. Blood Se levels can vary widely in humans, and this can be attributed to the Se levels in diets. The Se levels in blood plasma tend to indicate short-term Se status, and levels in red blood cells are indicative of long-term Se status. In blood, the Se organic forms can be oxidised to selenite or selenate and taken up by the red blood cells for delivery to the liver for metabolism and selenoproteins synthesis [82]. The levels of Se in blood plasma are consistently correlated with Se daily intake [83]. The typical levels of Se blood in a healthy population from selected countries are shown in Table 5.

Combs has provided a comprehensive review of Se biomarkers for tissue levels in blood, urine, nails and hair, and selenoproteins functions for assessing Se deficiency diseases and adverse health effects from excessive intake in humans [95]. For very high Se exposure, the biomarkers of blood plasma Se levels less than 1000 μg/L and Se intakes less than 1500 μg/day have been reported to cause no adverse health effects in human subjects [95]. In the past few years, the levels of selenoprotein P, which is predominantly found in the blood plasma, has increasingly been used for assessing the Se status in a population. Plasma selenoprotein P has been recommended as a suitable biomarker for assessing Se adequacy because it reaches a plateau at optimal levels of about 65 μg Se/L when Se intake is about 93 μg/day and over in a population [83,96]. Glutathione peroxidase (GPx) activity in blood plasma has historically been used as a biomarker in assessing Se adequacy in humans; however, because it reaches a plateau when Se intake is about 35 μg/day, it is a less reliable biomarker than selenoprotein P [83,97]. Low selenoprotein P levels in blood as a biomarker could link to health risks, and at high levels it could provide information about possible Se toxicity (selenosis) [98].

The levels of Hg in blood are widely used to assess Hg exposure in a population because Hg is well distributed via the blood to various target organs and tissues, and is considered a reliable biomarker that can reflect both elemental and methyl Hg exposures [99]. Other biomarkers for assessing the risks of exposure from Hg toxicity and adverse health effects in humans have been comprehensively reviewed [99,100]. Table 6 shows the blood Hg levels of healthy populations from selected countries.

Exposure to Hg in a work environment is the result of inhaling air pollutants and ingesting contaminated food and water. High blood Hg has been reported in workers who were exposed to Hg vapours and consumed food near Hg mining sites [15,112,113]. In countries such as Japan, Korea and China, the high blood Hg levels have been reported to be associated with a high consumption of fish and seafood products [22,28,114]. In Japan, the Hg level in whole blood has been reported to reach 30.6 µg/L after consumption of fish and seafood [106]. High blood Hg levels (range: 0.01–241 µg/L) have also been reported in the Inuit population, whose diet is mainly marine fish and other aquatic animals [104,115]. Considerably higher blood Hg levels have also been shown in Brazilian populations from fish consumption, where the total Hg levels in plasma ranged from 2.4 to 27.3 µg/L, and 8.4 to 83.2 µg/L for whole blood [102]. A long-term fish and shellfish intake increased the blood Hg to 17 µg/L in young children and it has been reported to cause a reduction in mental ability and concentration or attention [116]. In populations where fish and seafood products are not considered as a major diet component, blood Hg levels are generally lower, and this is shown in the low blood Hg levels of the Native American population [111]. There have been a few studies on Se biomarkers that could be used to assess excessive Hg exposure from co-intakes of Se and Hg in humans. In one study, the healthy subjects (non-exposed to elemental Hg) exposed to high methyl Hg from fish consumption showed a positive correction between levels of blood Hg and plasma selenoprotein P, but not GPx, which suggests the sensitivity of selenoprotein P to Hg exposure and its demand for supplies to various organs [23]. In another study, high consumption of fish or seafood as co-dietary intakes of Se and Hg in healthy populations showed correlation between whole blood Hg and plasma Se, and there was also a correlation between dietary intakes and whole blood Hg, but not with whole blood Se or plasma Se, which indicated that the correlation was not the result of co-intakes of the two elements from seafood [117]. The inconsistency of Se status as biomarkers to assess Hg exposure could possibly be explained by the complex presence of Se molecular species in biological tissues, and, unlike Hg, Se can come from various dietary sources that could contribute to different effects on Hg exposure outcomes. For instance, different fish species have been shown to contain different forms of Se compounds, and in some fish, Se was found only as selenomethylselenocysteine (SeMeSeCys) in different protein fractions in muscle tissue; this could play a significant role in Se-Hg interactions and the effect of Hg exposure [118].

There has been increasing concern about Hg exposure in high-risk groups such as pregnant women, infants and young children. Numerous studies from various countries have assessed the Hg status in pregnant women by determining the Hg levels in blood and umbilical cord blood after birth [119,120,121]. Maternal consumption of fish has been shown to increase Hg levels in cord blood, and these levels were significantly higher than other maternal groups who did not consume fish [120]. Maternal consumption of large amounts of marine fish has been shown to be associated with increased levels of Hg in cord blood, which could result in the accumulation of Hg in infants [122]. The elevated levels of Hg in maternal blood after fish consumption have also been reported to be significantly correlated with cord blood Hg, and could pose risk to foetal development [123]. The neurodevelopmental effects of Hg exposure on infants and young children are of particular concern because of their greater vulnerability to toxicity. A number of endpoints have been investigated to determine the association between Hg exposure and neurodevelopmental effects, and these include neurocognitive, motoric and auditory, visual electrophysiological responses and neurobehavioral effects [124]. In the follow-up studies of children who were mainly on fish diets as part of the Seychelles Child Development Study, the findings did not reveal any consistent association between prenatal Hg exposure and neurodevelopment endpoints [69]. A recent study of methyl Hg exposure from fish diet in young children has shown that there was an association between high prenatal methyl Hg exposure and differential DNA methylation in children’s saliva at seven years of age at specific CpG sites of the nervous system-related genes that may influence neurodevelopment and mental health [125]. This recent finding of DNA methylation in nervous system-related genes from Hg exposure may provide further evidence of mechanisms of developmental neurotoxicity.

Concern about the increased Hg exposure in populations has caused government regulatory bodies in many countries to provide information on strategies to reduce exposure from ingesting contaminated food and water, and inhaling polluted air. Over the years there have been declining trends of blood Hg levels in some countries, but in others the blood levels are increasing, and this is due to increased industrial development in these countries [100,126].

## 6. Se and Hg Interactions and Health Implications

The consumption of fish and seafood is a major dietary source of protein in many parts of the world, and this could also be a major contributor to essential nutrients and contaminants including essential Se and the toxic metal Hg. There has been increasing interest in the study of interactions between Se and Hg since it was observed that Se could reduce Hg toxicity by increasing its elimination, and there are significant correlations between these two elements in fish and other seafood tissues [127,128]. Each form of Se compound is effective in reducing Hg toxicity by forming inert Se–Hg complexes in biological tissues; however, this effectiveness is also dependent on the Se and Hg chemical forms. For instance, selenite, selenomethionine and selenocysteine can have similar effects in eliminating methyl Hg, but not selenate, to reduce Hg toxicity in shrimps [129]. However, selenate and selenocysteine are more effective than selenomethionine and selenite in reducing the toxicity of inorganic Hg (II) by elimination in fish [13]. In rat studies, the effectiveness of Se compounds in reducing Hg toxicity can generally be presented in the order of greater to lesser effectiveness as: selenomethionine > selenocysteine > selenate > selenite [130,131].

Even though there have been numerous studies on Se–Hg interactions in animals, these studies have not been clearly demonstrated in humans and these may have health implications such as the effects of Hg on cancer, neurodevelopment and immunity [68,132,133,134,135]. However, human cell lines have been used to investigate the effectiveness of Se compounds on the toxicity of Hg compounds that specifically targeted the thioredoxin systems for mechanisms of toxicity [99,136].

It has been demonstrated in animal studies that both Hg (II) and methyl Hg can cause oxidative stress from the formation of reactive oxygen species (ROS) in biological tissues, and the presence of Se can remove the effects of ROS through its roles in antioxidant selenoenzymes such as glutathione peroxidase (GPx) and thioredoxin reductase (TrxR) of the thioredoxin (Trx) systems [137,138]. The production of ROS after exposure to mercury chloride (HgCl_2_) has been demonstrated in human endometrial stromal cells which caused decreases in cell proliferation and promoted apoptosis and could be linked to human infertility [139]. Both the glutathione and thioredoxin systems are the targets of Hg compounds, and it has been shown that elevated Hg levels in the form of ethyl mercury in human neuroblastoma cell lines caused depletion of GPx and TrxR due to oxidation, leading to caspase activation and apoptosis [140]. In neuroblastoma cell lines, the presence of methyl Hg affected TrxR more than GPx, but the addition of Se compounds in the form of diphenyl diselenide ((PhSe)_2_) and ebselen helped to protect and restore the activities of these selenoenzymes [141]. The potential actions of these Se compounds against Hg toxicity include their ability to scavenge ROS, and in particular (PhSe)_2_, but not ebselen, to induce the synthesis of TrxR for maintaining cell homeostasis [138,141].

The proposed mechanisms of Hg toxicity and its attack on the integrity of the selenoenzymes, particularly mammalian Trx systems, which consist of mammalian cytosolic Trx1 and mitochondrial Trx2, have attracted increasing attention. The Trx network is very sensitive to the exposure of Hg compounds because it contains dithiol active sites (-Cys-Gly-Pro-Cys-) and three additional cysteine residues that have binding affinity for Hg compounds, and this could lead to a loss of catalytic activity [142,143]. The binding of Hg to sulphur at the cysteine active site has been suggested as the mechanism of Hg toxicity as a result of S-Hg bonds (methyl-Hg-S-Cys, Hg-S-Cys) that would alter the protein structure in tissues [128,143,144,145]. Like sulphur, the presence of selenocysteine as an active site in TrxR, and its attack by Hg compounds, could also lead to the formation of Se–Hg bonds. However, by comparing the binding affinity of sulphur with Hg in physiological conditions, Se (log k = 45) is millions of times greater than S (log k = 39) [18,33].

The strong binding formation of Se-Hg in biological systems has been suggested as a basis of the Se protective effect against Hg toxicity. By forming inert methyl-Hg-Se-R and Hg-Se complexes, Hg is distributed to various tissues for elimination and may reduce its toxicity, but this could also lead to Se deficiency, decreased selenoprotein levels and decreased selenoenzyme activities [120]. The formation of methyl-Hg complexes with S or Se has been proposed for the mechanisms of Hg toxicity in the brain as it passes through the blood-brain-barrier [146]. Se is particularly important to brain function, and when Se is depleted in other tissues, it is maintained in the brain for homeostasis [133,147]. Severe Se deficiency could cause irreversible brain injury from Hg toxicity, and the preserved expression of two thioredoxin reductases (cytoplasmic TrxR1 and mitochondrial TrxR2) in the brain could also play an important role in protection against chronic Hg exposure [18,137]. Hg compounds (inorganic and organic Hg) have been shown to cross the blood-placental barrier in several studies, both in vivo and in vitro; however, the specific molecular mechanisms of Hg crossing the placental barrier are not fully understood [148]. In a pregnant rat study, the exposure to inorganic Hg (HgCl_2_) caused an increase in Hg levels in placental and foetal organs, which were also dependent on the time and the dose of exposure [149]. In a recent study of pregnant Japanese women, it was observed that methyl Hg levels in the red blood cells and plasma of cord blood were significantly higher than in maternal blood, suggesting that some inorganic Hg was trapped in the placenta and that methyl Hg in cord blood could be easily absorbed by the foetal brain [150].

It has been suggested that the demethylation of Hg compounds with the formation of inert Hg-Se complexes, and the binding of Hg-Se into selenoprotein P (SelP) structure, could also be the mechanistic actions of Se protective effects in tissues [18,133,151]. In fish, after being exposed to methyl Hg and treated with different dietary Se levels, this has caused a significant increase in the demethylation of methyl Hg and the reduction of its accumulation in tissues [151,152]. In animal studies, treatment with a Se-rich diet has caused a reduction of Hg accumulation levels in brain, liver and kidney [16,153]. Increased dietary Se intake in mice treated with methyl Hg has been shown to reduce the accumulation of methyl Hg in target tissues and prevent redox-mediated immunosuppression [134]. These animal studies seem to provide evidence that the presence of Se at certain levels during Hg exposure reduces the distribution and accumulation of Hg in tissues, and thus reduces Hg toxicity. However, a recent study has shown that mice co-administrated intragastrically with Hg (methyl Hg or inorganic Hg) and Se (selenite) developed a significant increase in Hg levels in tissues (brain, kidney, liver, spleen) compared to levels administered only with Hg, indicating that Se could promote the absorption of Hg into tissues [154].

The Hg toxic effects in these tissues have been shown to be associated with the concentration of a molar ratio of Se:Hg, and if the molar ratio value is greater than one, this would indicate a greater amount of Se for protection against Hg toxicity, particularly with respect to the concentration of Se and Hg in fish [155]. The molar ratio of Se:Hg is based on the calculation of molar concentration of total Se divided by molar concentration of total Hg. This Se:Hg molar ratio has been proposed as a useful criterion to assess Hg toxicity in blood and other tissues, and to determine the relationship between methyl Hg (total Hg) and toxicity [153]. High values of Se:Hg molar ratios (range: 5–625) for Se and Hg concentrations in human placenta have been used to assess the potential of Hg toxicity; however, the basis of the molar ratio criterion is questionable because Se can bind to other cations and different forms of Hg compounds in tissues [121,128].

In a population where the principal dietary source of protein is fish and seafood, the Se:Hg molar ratio has been used as a guideline to assess the risk of Hg toxicity. In Norway, a comprehensive survey of marine fish species has shown a wide variation of Se and Hg concentrations, with Se:Hg molar ratios greater than 1, ranging from 1.5 to 51.1 for all fish species, indicating a low risk of Hg toxicity [156]. Freshwater fish species have also been shown to contain low levels of Hg, and because of high Se:Hg molar ratio values, the serving of freshwater fish meals per week has been advised to women, particularly for women of childbearing age [157].

Even though the Se:Hg molar ratio has been useful as a guideline for assessing the risk of consuming fish and seafood, it has not been included in any policy of advisory or regulatory bodies on the recommendation for required safe intake of fish and seafood. In Europe, EFSA has suggested that each country has to assess and advise its own population on the recommendation of adequate fish and seafood intake because there is a significant variation of dietary patterns of fish and seafood intake between countries [158]. However, because fish can be important sources of essential nutrients such as omega-3 fatty acids, alphalinolecic acid (ALA), eicosapentaenoic (EPA), and docosahexaenoic acid (DHA), various countries have provided guidelines for the amount of fish intake for a population. There is also a variation on the recommendation of fish and seafood intake between countries [159]. For instance, in Australia, the recommendation for pregnant women is 150 g (Orange Roughy fish) per week or 150 g of shark flake per fortnight [160]. In the USA, the intake of Orange Roughy or shark is not recommended for women of childbearing age [161]. For the UK, pregnant women are recommended to take 170 g of raw tuna or 140 g of cooked weight per week, but limit intake of oily fish [159].

## 7. Concluding Remarks

Insights into further understanding the mechanisms of Se and Hg interactions at a molecular level are important if the effects of Hg toxicity can be alleviated by specific selenium compounds at their optimal concentration required for the protection of target tissues. This area of investigation remains a major challenge because of the complexity of these interactions, involving the competition and transportation of Se and Hg compounds (inorganic and organic compounds) and subsequent transformation in biological tissues [148]. The long-term effects of low dose Hg exposure and the health implications are still unclear, particularly the effects on pregnant women and their infants. A recent study on gene expression profiles from low dose Hg exposure could affect the functions of autophagy of mitochondrion and oxidative stress, and may alter genes for immune function in pregnant women and newborns that could lead to negative effects on fine motor skills in young children [162,163]. Until there is new insight on biomarkers of Hg toxicity from the long-term exposure, it is important that the at-risk groups in a population should avoid or minimise Hg exposure and ensure that adequate daily intake of Se is maintained.

## Figures and Tables

**Table 1 nutrients-14-05308-t001:** Typical Selenium levels in food from selected countries.

Food	Selenium Level (µg/kg)	Country	Reference
Milk and dairy products			
Milk	10.7–16.2	Greece	[35]
Milk	10.0–14.0	Switzerland	[36]
Milk	60.0	Korea	[37]
Milk	22.5–25.9	Australia	[38]
Cheese	24.1–95.4	Greece	[35]
Cheese	70.0–78.9	Australia	[38]
Meat and eggs			
Beef	33.5–6.31	Greece	[35]
Beef	67 ± 23	Switzerland	[36]
Beef	324	Korea	[37]
Lamb	80	Switzerland	[36]
Chicken	76.3–82.4	Greece	[35]
Chicken	114 ± 17	Switzerland	[36]
Pork	90.0–98.2	Greece	[35]
Pork	115 ± 75	Switzerland	[36]
Pork	174–199	Korea	[37]
Eggs	56.4–181.1	Greece	[35]
Eggs	190–414	Australia	[38]
Cereals			
Bread	37.9–150.2	Greece	[35]
Bread	23–48	Switzerland	[36]
Bread	216	Korea	[37]
Bread	92.6–125	Australia	[38]
Rice	17.7–20.5	Greece	[35]
Rice	50	Korea	[37]
Rice	25	Australia	[38]
Fruit and vegetables			
Apple	1.1–1.9	Greece	[35]
Banana	4.3–5.7	Greece	[35]
Pear	4.7–7.9	Greece	[35]
Broccoli	6.1–11.8	Greece	[35]
Broccoli	6	Korea	[37]
Carrot	3.6–8.5	Greece	[35]
Potato	3.1–6.0	Greece	[35]
Fish and seafood			
Sardine	261.5–332.8	Greece	[35]
Trout	28.7–96.9	Greece	[35]
White fish	210 ± 58	Switzerland	[36]
Perch	303 ± 46	Switzerland	[36]
Tuna	659	Korea	[37]
Shrimp	251	Korea	[37]
Fish	120–632	Australia	[38]

**Table 2 nutrients-14-05308-t002:** Selenium daily intake from selected countries.

Se Intake (µg/day)	Age Group	Population Group	Country	Reference
67–90	Adults	Men	Australia	[51]
52–57	Adults	Women	Australia	[51]
59.6	Adults	all	Belgium	[52]
44	Adults	all	China	[53]
60	Adults	Men	Finland	[40]
50	Adults	Women	Finland	[40]
52 ± 14	Adults	all	France	[54]
39.3	Adults	all	Greece	[35]
98	Adults	all	Japan	[55]
57.5	Adults	all	Korea	[37]
51 ± 26	Adults	all	New Zealand	[56]
60 ± 25	Adults	Men	New Zealand	[56]
46 ± 26	Adults	Women	New Zealand	[56]
50.4	Adults	all	Switzerland	[36]
48 ± 14	Adults	all	UK	[57]
54 ± 15	Adults	Men	UK	[57]
43 ± 11	Adults	Women	UK	[57]
92.1 ± 42.6	Adults	all	USA	[58]

**Table 3 nutrients-14-05308-t003:** Typical mercury levels in food from selected countries.

Food	Mercury Level (µg/kg)	Country	Reference
Milk and dairy products			
Cheese	<0.27–1.8	Canada	[62]
Cheese	0.37 ± 0.15	Spain	[63]
Milk	0.25 ± 0.06	Spain	[63]
Milk	<0.13–0.25	Canada	[62]
Meat and eggs			
Eggs	0.45 ± 0.16	Spain	[63]
Eggs	0.39–1.5	Canada	[62]
Beef	0.42–1.8	Canada	[62]
Lamb	0.29–2.3	Canada	[62]
Pork	0.68–1.9	Canada	[62]
Red meat	0.54 ± 0.19	Spain	[63]
Fish and seafood			
Cuttlefish	34	Spain	[64]
Fish (freshwater)	69–83	Canada	[62]
Fish canned	63–148	Canada	[62]
Prawn	91	Spain	[64]
Sardine (fresh)	80	Spain	[64]
Squid	52	Spain	[64]
Swordfish	1500	Spain	[64]
Tuna (fresh)	680	Spain	[64]
Cereals and grains			
Wheat and bran	<0.45–1.4	Canada	[62]
Bread	0.14–0.37	Canada	[62]
Rice	0.57–1.8	Canada	[62]
Vegetable and fruit			
Bean	<0.12–0.22	Canada	[62]
Broccoli	0.40–0.67	Canada	[62]
Corn	<0.11–0.21	Canada	[62]
Potatoes	<0.08–0.16	Canada	[62]

**Table 4 nutrients-14-05308-t004:** Total mercury intake from selected country.

Intake (µg/day)	Age Group	Population Group	Country	Reference
4.6–12.9	Adults	all	Australia	[70]
0.77–2.17 *	Adults	all	Canada	[62]
5	Adults	all	Chile	[71]
5.74 (2.87–12.9) *	Adults	Men	China	[72]
5.27 (2.34–10.55) **	Adults	Women	China	[72]
9.65	Adults	all	France	[73]
0.5	Adults	all	India	[74]
15.3 (2.65–48.4) ***	Adults	Women	Japan	[75]
3.8 ± 5.26	Adults	all	Korea	[76]
1.61	Adults	all	Korea	[77]
6.65 ± 2.01 (3.2–10.7)	Adults	Men	Poland	[78]
5.21 ± 1.56 (2.9–9.5)	Adults	Women	Poland	[78]
5.57	Adults	all	Spain	[63]
7.45 *	Adults	Men	Spain	[79]
7.0 *	Adults	Women	Spain	[79]
1.0–3.0	Adults	all	UK	[80]
1.25–2.59	Adults	Men	USA	[81]
0.97–1.95	Adults	Women	USA	[81]

* An asterisk indicates conversion of body weight intake (µg/Kg bw/day) to µg/day. ** Closed brackets indicate range values. *** Values are expressed in geometric mean (GM).

**Table 5 nutrients-14-05308-t005:** Selenium levels in blood from selected countries.

Blood Se Levels (µg/L)	Blood Matrix	Population Group	Country	Reference
141 (118–224)	Whole blood	all	Australia	[84]
142 (118–224)	Whole blood	Men	Australia	[84]
140 (122–203)	Whole blood	Women	Australia	[84]
130 (82–180)	Plasma	all	Australia	[84]
130 (101–161)	Plasma	Men	Australia	[84]
124 (82–179)	Plasma	Women	Australia	[84]
163.0 * (123–205)	Whole blood	all	Benin	[85]
89.3 (68–245)	Whole blood	all	Brazil	[86]
109.1 (949.5–195.2)	Whole blood	Men	Finland	[87]
97.9 (30.0–244.8)	Whole blood	Women	Finland	[87]
107 (75–146)	Whole blood	all	Germany	[88]
32–178	Whole blood	all	India	[89]
35.8–185.6	Serum	all	India	[89]
111.4 ** (79.1–166.5)	Serum	all	Korea	[90]
55.3–207.4	Serum	Men	Korea	[90]
70.4–171.7	Serum	Women	Korea	[90]
111.5	Serum	Men	New Zealand	[91]
66.3 (63.0–71.2)	Serum	Men	Serbia	[92]
63.8 (61.9–70.2)	Serum	Women	Serbia	[92]
115 * (60.3–226)	Whole blood	Men	Slovenia	[93]
94.6 * (53.9–176)	Whole blood	Women	Slovenia	[93]
89.2 ± 12.6	Plasma	all	UK	[57]
91.6 ± 12.6	Plasma	Men	UK	[57]
87.6 ± 13.4	Plasma	Women	UK	[57]
146.8-247.3 ***	Whole blood	all	USA	[94]
149.4–250.4 ***	Whole blood	Men	USA	[94]
144.2–244.6 ***	Whole blood	Women	USA	[94]

* Values are expressed as geometric mean (GM) and range from 5% to 95% percentiles. ** Values are expressed as geometric mean (GM) and range from 2.5% to 97.5% percentiles. *** Range values 2.5% to 97.5% percentiles.

**Table 6 nutrients-14-05308-t006:** Mercury levels in blood from selected country.

Blood Hg Levels (µg/L)	Blood Matrix	Population Group	Country	Reference
0.83 (<0.2–5.80)	Whole blood	Pregnant women	Australia	[101]
2.0 (<0.8–9.3)	Whole blood	All	Australia	[84]
2.1 (<0.8–7.7)	Whole blood	Men	Australia	[84]
1.8 (<0.8–9.3)	Whole blood	Women	Australia	[84]
3.12 (1.11–7.64) *	Whole blood	All	Benin	[85]
8.4–83.2	Whole blood	All	Brazil	[102]
9.6 (2.4–27.3)	Plasma	All	Brazil	[102]
1.4 (0.10–12.40)	Whole blood	All	Brazil	[103]
21.1 ± 24.7	Whole blood	All	Denmark	[104]
0.65 (0.03–3.5)	Whole blood	All	Germany	[88]
0.2 (<0.02–1.1)	Serum	All	Germany	[88]
5.11 ± 2.19 (1.16–15.79)	Whole blood	Children (M and F)	Japan	[105]
4.41 (0.35–30.6)	Whole blood	All	Japan	[106]
3.12–5.66 *	Whole blood	Men	Korea	[107]
2.45–3.85 *	Whole blood	Women	Korea	[107]
3.12 (2.96–3.28)	Whole blood	All	Korea	[76]
4.94 (4.66–3.53) **	Whole blood	Men	Korea	[108]
3.27 (3.13–3.42)	Whole blood	Women	Korea	[108]
1.3 (0.39–4.4)	Whole blood	Men	Sweden	[109]
0.97 (0.17–2.9)	Whole blood	Women	Sweden	[109]
0.77 (0.71–0.83)	Whole blood	Adolescents (M)	Sweden	[110]
0.60 (0.56–0.64)	Whole blood	Adolescents (W)	Sweden	[110]
0.359 *	Whole blood	All	USA	[111]

* Values are expressed in geometric mean (GM) with range values in closed brackets from 25% to 95% percentiles. ** Range values in closed brackets are expressed in confidence interval (CI).

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
