# Peer review of "Selenium Status: Its Interactions with Dietary Mercury Exposure and Implications in Human Health"

_nutrients, 2022, doi:10.3390/nu14245308_

Round 1

Reviewer 1 Report

Selenium is an essential trace element with a well-documented role in selenoprotein and enzyme antioxidant activity. Food is the principal source of selenium.  There is extensive research on the roles of selenium in mitigating the toxic effects of mercury exposure from human dietary intake. The aim of this review is to summarise current information on the interplay of the interactions between selenium and mercury in the body and the protective effect of selenium on at-risk groups in a population that may experience long-term mercury exposure. This area of investigation remains a big challenge due to the complexity of selenium-mercury interactions. The long-term effect of low-dose mercury exposure and its negative effects on health still require additional research. This manuscript reviews 145 articles. The topic of this manuscript is up-to-date, attractive and well-suited for the Journal Nutrients. The manuscript is well-written and divided into seven parts. The text is clear and easy to read.  The authors used 6 tables. This aid the readers' understanding. I suggest checking for some minor spelling mistakes and grammar errors. Otherwise, I  have no major concerns regarding this manuscript.

Author Response

I would like to thank the reviewer for comments on the requirement of spelling check and grammatical corrections. These comments have been addressed throughout the manuscript.

Reviewer 2 Report

The authors have conducted a narrative review of dietary sources and blood levels of Se and Hg and discussed the interaction of various Se- and Hg-compounds. The authors outline the complexities of the interactions between Se and Hg as it relates to human health.

Primary comments:

11)      I recommend organizing this review around more specific research/review question or questions that this review can provide a novel summary or overview of. As currently written, the review touches on material stated in the goals (focused on the interplay of Se and Hg). To me, the studies and data reported in the tables only report on Hg and Se separately for dietary sources and blood levels. It seems that the forms of Se and Hg is a primary theme, and if that is the authors’ intention, results should be organized considering the forms and included throughout (e.g. different dietary forms, forms as biomarkers, etc)

Based on the stated aims, I would look for reorganized/expanded sections on:

a.       different chemical forms of Hg and Se as it relates to absorption, digestion, metabolism, excretion, (patho)physiological function.

b.       To parallel a, it may be helpful to have an overview figure showing how the different forms of Hg and Se are present in the environment, dietary exposures, blood, tissues, and how these forms interact with each other.

c.       If the goal is to discuss human health outcomes, I recommend explicitly stating what human health outcomes you intend to review, then organize sections by health outcome that include evidence of Hg/Se in relation to that outcome. I would organize this separately by cell/animal studies and then human studies to organize the evidence base for the reader.

d.       Conclusion section should summarize overall answers provided to research questions, and list more specific research directions that would be beneficial.

2)    Please add methods of how this review was conducted.  Was any literature search performed? Please include the relevant search terms and methods used. It is important to know whether the results presented represent a systematic overview, or are a convenient sample.

 Additional comments:

L28-29: Is there available evidence of Hg and Se bioaccumulation that can be cited here?

L 114: Can you add why methyl Hg is the most toxic? (this may be more appropriate in another section as well)

L138: I would recommend expanding this section or having a separate section on nutrient reference values for Se. What is known about absorption, digestion, metabolism, excretion as it relates to different forms of Se. What are estimated requirements by life stage, what are estimated levels of toxic exposure, and what is the recommended safe upper limit of exposure?

Recommend also bringing in the need to avoid Se toxicity as well, based on different forms.

You have some more in L 147-150. What is the basis for these recommended intakes? 

L156: what are these elevated concentrations in rice near mining areas? How much would this expose somebody consuming rice as a staple food?

Section 5:

Please expand on the use of blood concentrations as a biomarker – recommend discussing this in relation to Se metabolism and storage to justify how closely blood levels reflect amounts in tissues.

I see more in L179-186. How do these biomarkers relate to toxicity as well? Is there an upper limit or different biomarker?

Table 5: Please clarify differently expressed values. * look like they are used in the legend but I don’t see these in the table. It also looks like there are some values with +/-, are these SDs? Some rows have ‘GM’ but all values are listed as geometric mean?

Table 6: Recommend changing ‘sex’ column to population group, etc. or have separate columns for age, sex, pregnancy.

L238: This paragraph seems a bit disconnected.

L291: is there any evidence more than cord blood linking maternal Hg to infant exposure/status?

L294-296 please clarify direction of association between dietary Se and Hg outcomes.

L301: ‘concentration of molar-ratio’ I don’t understand this sentence, please clarify

 L303:308 – recommend putting in rationale for biomarkers section

L312: does the Se:Hg molar ratio depend on the forms of Se and Hg?

L328: recommend replacing ‘fortnight’ with other expression of time

For Table 2, are these healthy populations?

Line 340-343: Recommend putting this in a relevant review section and restructuring the conclusion section for overall conclusions and research recommendations.

I found this on a quick search that may be relevant:

Liu, et al. 2021  Analytica Chemica Acta “Investigation on selenium and mercury interactions and the distribution patterns in mice organs with LA-ICP-MS imaging”

Author Response

We would like to thank you for your time of reviewing the manuscript. Your comments and suggestions have been very helpful to improve the manuscript. Details of our responses to your comments are presented:

Manuscript ID: nutrients-1984980

Type of manuscript: Review

Title: Selenium status: its interactions with dietary mercury exposure and implications in human health

Authors: Ujang Tinggi and Anthony V. Perkins

Response to reviewer 2:

We would like to thank the reviewer for the time to carefully read and provide comments and suggestions, and to help us to further improve the manuscript. We have carefully considered the comments and amended the manuscript accordingly. Matters raised are addressed on a point-by-point basis below. Our responses are shown in blue, and the reference to amended text is indicated by the line numbers in closed brackets.

1)  I recommend organizing this review around more specific research/review question or questions that this review can provide a novel summary or overview of. As currently written, the review touches on material stated in the goals (focused on the interplay of Se and Hg). To me, the studies and data reported in the tables only report on Hg and Se separately for dietary sources and blood levels. It seems that the forms of Se and Hg is a primary theme, and if that is the authors’ intention, results should be organized considering the forms and included throughout (e.g. different dietary forms, forms as biomarkers, etc)

Response: Thank you for the comments and suggestions. The manuscript objectives were mainly to provide overview and summary of current literature on information of Se status and Hg exposure through human diet. This review discusses Se and Hg in tandem, and separation into sub-headings on each chemical form of Se or Hg may be not necessary; however, we agree that the discussion should focus also on major forms of Se in biological materials, particularly food, which are present mainly as organic Se compounds such as selenoscysteine and selenomethionine. This is discussed in section of 4 Se status and Hg exposure. The discussion on the use of selenomethione, which is most bioavailable form, in food supplements has also been added in the text (lines 131-137).

Based on the stated aims, I would look for reorganized/expanded sections on:

  1. different chemical forms of Hg and Se as it relates to absorption, digestion, metabolism, excretion, (patho)physiological function.

Response: Thank you for the suggestions. We have provided and extended the discussion to include absorption, digestion, metabolism and excretion of Se from the body (lines 142-144).

  1. To parallel a, it may be helpful to have an overview figure showing how the different forms of Hg and Se are present in the environment, dietary exposures, blood, tissues, and how these forms interact with each other.

Response: Thank you for the suggestion. By providing overview schematic figures on how different forms of Hg and Se are being transformed would be useful presentation. However, we have added discussion on Se transformation in food and blood (lines 142-144, 1779-181), and bio-accumulation of methyl Hg in rice from Hg-contaminated area (lines 161-164).

  1. If the goal is to discuss human health outcomes, I recommend explicitly stating what human health outcomes you intend to review, then organize sections by health outcome that include evidence of Hg/Se in relation to that outcome. I would organize this separately by cell/animal studies and then human studies to organize the evidence base for the reader.

Response: Thank you for the suggestions. The paper includes discussion on human health implications when expose to both Se and Hg, but it does not emphasise the discussion on human health outcomes relating to data evaluation, evidence-based data for intervention and disease risk. The paper does discuss the use of blood levels as biomarkers for assessing elevated levels of Se and Hg from dietary exposure.

  1. Conclusion section should summarize overall answers provided to research questions, and list more specific research directions that would be beneficial.

Response: The paper does not propose to provide research questions as it is not designed for systematic literature search for determining or evaluating health outcomes from Se and Hg dietary exposures.

2)    Please add methods of how this review was conducted.  Was any literature search performed? Please include the relevant search terms and methods used. It is important to know whether the results presented represent a systematic overview, or are a convenient sample.

Response: The literature was performed on PubMed and ScienceDirect. This manuscript was organised to undertake general overview of current literature on Se status and Hg exposure through food relating to human health, and a summary of blood levels which are commonly used for biomarkers of dietary exposure. The manuscript was not aimed to conduct comprehensive systematic review, and no methodology was selected for data extraction and evaluation for assessing health outcomes.

 Additional comments:

L28-29: Is there available evidence of Hg and Se bioaccumulation that can be cited here?

Response: the references have been added (line 29).

L 114: Can you add why methyl Hg is the most toxic? (this may be more appropriate in another section as well)

Response: Methyl Hg is highly toxic because of its ability to disrupt protein function by forming low molecular weight with thiol, and this discussion has been added in the text with reference (lines 107-109).

L138: I would recommend expanding this section or having a separate section on nutrient reference values for Se. What is known about absorption, digestion, metabolism, excretion as it relates to different forms of Se. What are estimated requirements by life stage, what are estimated levels of toxic exposure, and what is the recommended safe upper limit of exposure?

Response: Thank you for the suggestions for a separate section. However, we would rather expand the discussion on absorption, digestion, metabolism and excretion for different forms of Se and their toxic effects. This discussion has been added in the text (lines 142-144; 180-182) including the relevant references.

Recommend also bringing in the need to avoid Se toxicity as well, based on different forms.

Response: This discussion has been added with references (lines 131-137; 148-153).

You have some more in L 147-150. What is the basis for these recommended intakes? 

Response: The basis of these recommended intakes is to provide guideline and reference values for adequate Se intakes for a population in some countries including Australia. This discussion has been added in the text (148-153).

L156: what are these elevated concentrations in rice near mining areas? How much would this expose somebody consuming rice as a staple food?

Response: The data and discussion on elevated Hg levels in rice near mining areas have been added (lines 161-164).

Section 5:

Please expand on the use of blood concentrations as a biomarker – recommend discussing this in relation to Se metabolism and storage to justify how closely blood levels reflect amounts in tissues.

Response: The discussion has been added in the text (lines 175-181).

I see more in L179-186. How do these biomarkers relate to toxicity as well? Is there an upper limit or different biomarker?

Response: Se intake/status biomarkers relating to toxicity can be presented with U-shaped response, and this is discussed in the text (lines 145-153).

Table 5: Please clarify differently expressed values. * look like they are used in the legend but I don’t see these in the table. It also looks like there are some values with +/-, are these SDs? Some rows have ‘GM’ but all values are listed as geometric mean?

 Response: Corrections have been made and provided by the footnotes of Table 5.

Table 6: Recommend changing ‘sex’ column to population group, etc. or have separate columns for age, sex, pregnancy.

Response: Correction has been made by changing “sex” to “population group”.

L238: This paragraph seems a bit disconnected.

Response: The paragraph has been separated and added with further discussion (lines 323-325).

L291: is there any evidence more than cord blood linking maternal Hg to infant exposure/status?

Response: Hg in breast milk could also contribute to infant exposure, and this discussion has been added in the text (lines 223-225).

L294-296 please clarify direction of association between dietary Se and Hg outcomes.

Response: the sentence has been re-written for clarity (see lines 321-325).

L301: ‘concentration of molar-ratio’ I don’t understand this sentence, please clarify

Response: The sentence has been re-written for clarity (lines 321-325).

 L303:308 – recommend putting in rationale for biomarkers section

Response: The molar ratio is used as a reference for comparison of different concentrations between Hg and Se that has been used as a guideline for risk assessment, but not as biomarker assessment for health outcomes. The sentence has been re-written (lines 321-325).

L312: does the Se:Hg molar ratio depend on the forms of Se and Hg?

Response: The molar ratio value is used to determine the total levels of Se and Hg particularly for levels in fish, which are predominantly found as organic compounds, and thus does not entirely depend on chemical forms for evaluation. These molar ratio values are then used as a guideline for assessing the potential risk of Hg toxicity.

L328: recommend replacing ‘fortnight’ with other expression of time

Response: the sentence has been re-written to replace ‘fortnight’ (lines

For Table 2, are these healthy populations?

Response: Table 2 represents data of healthy population and this has been included in the text (line144-145).

Line 340-343: Recommend putting this in a relevant review section and restructuring the conclusion section for overall conclusions and research recommendations.

I found this on a quick search that may be relevant:

Liu, et al. 2021  Analytica Chemica Acta “Investigation on selenium and mercury interactions and the distribution patterns in mice organs with LA-ICP-MS imaging”

Response: Thank you for the literature which is very relevant to this manuscript. We have included this information for discussion in the main text (lines 313-319).

Reviewer 3 Report

The review 'Selenium status: its interactions with dietary mercury exposure and implications in human health' provides basic information on the human exposure to Se and Hg and describes the potential health impact linked to their interaction.

The paper is quite well written, but some aspects of Se and Hg toxicological profiles are not sufficently debated.

First of all, since a review article should be comprehensive, I strongly suggest the Authors to mention also the data of the European Food Safety Authority concerning both Se and Hg. Indeed, several pivotal information on dietary exposure, source of exposure, levels in the foodstuff, etc. which concern exposure of European population are lacking. The Authors should consider EFSA journals for both Se and Hg.

Moreover, the neurodevelopmental toxicity of Hg is a key endpoint which should be carefully analyzed and mentioned.

I suggest a moderate revision of th English language to clarify several sentences mainly in the Introduction and in the Discussion.

Author Response

We would like to express our thanks for your time in reviewing the manuscript. Your comments and suggestions have helped us to improve the manuscript. Details of our responses to address our comments are presented:

Manuscript ID: nutrients-1984980

Type of manuscript: Review

Title: Selenium status: its interactions with dietary mercury exposure and implications in human health

Authors: Ujang Tinggi and Anthony V. Perkins

Response to reviewer 3:

We would like to thank the reviewer for the time to carefully read and provide comments and suggestions, and to help us to further improve the manuscript. We have carefully considered the comments and amended the manuscript accordingly. Our responses are shown in blue, and the reference to amended text is indicated by the line numbers in closed brackets.

  • First of all, since a review article should be comprehensive, I strongly suggest the Authors to mention also the data of the European Food Safety Authority concerning both Se and Hg. Indeed, several pivotal information on dietary exposure, source of exposure, levels in the foodstuff, etc. which concern exposure of European population are lacking. The Authors should consider EFSA journals for both Se and Hg.

  • Response: Thank you for the suggestions. We have added this information regarding EFSA data on Hg for the discussion in the text (lines 167-171), and data on Se (lines 131-137).

  • Moreover, the neurodevelopmental toxicity of Hg is a key endpoint which should be carefully analyzed and mentioned.

  • Respones: We have made chances and added further discussion on neurodevelopmental toxicity of Hg in the text, including relevant reference (lines 224-236).

  • I suggest a moderate revision of the English language to clarify several sentences mainly in the Introduction and in the Discussion.

  • Response: We have made appropriate changes to sentences for clarity throughout the manuscript.

Round 2

Reviewer 2 Report

Thank you authors for responding to review comments and providing minor revisions addressing some comments.

Remaining major items include:

#1 (Previous 1c): see reviewer's response below.

Original reviewer comment: If the goal is to discuss human health outcomes, I recommend explicitly stating what human health outcomes you intend to review, then organize sections by health outcome that include evidence of Hg/Se in relation to that outcome. I would organize this separately by cell/animal studies and then human studies to organize the evidence base for the reader.

Author Response: Thank you for the suggestions. The paper includes discussion on human health implications when expose to both Se and Hg, but it does not emphasise the discussion on human health outcomes relating to data evaluation, evidence-based data for intervention and disease risk. The paper does discuss the use of blood levels as biomarkers for assessing elevated levels of Se and Hg from dietary exposure.

Reviewer response: To me, the relation between blood concentrations as biomarkers and adverse health outcomes is critical for interpreting the link between dietary exposures to health outcomes, and its absence reduces the review quality. Do the authors have a rationale for excluding this? If data is limited, this would still be valuable to discuss as a needed research direction.

#2 (Previous 2) Please add stated methods to the manuscript and include what search terms were included for PubMed and ScienceDirect, including the date databases were searched and the total number of retrieved results by database.

Author Response

Response to reviewer 2 of the revised manuscript:

We would like to thank the reviewer for the time to provide useful comments and suggestions to improve the revised manuscript. We have made further amendments in response to these comments, and our responses are shown in blue including line numbers for reference in the text.

#1 (Previous 1c): see reviewer's response below.

Original reviewer comment: If the goal is to discuss human health outcomes, I recommend explicitly stating what human health outcomes you intend to review, then organize sections by health outcome that include evidence of Hg/Se in relation to that outcome. I would organize this separately by cell/animal studies and then human studies to organize the evidence base for the reader.

Author Response: Thank you for the suggestions. The paper includes discussion on human health implications when expose to both Se and Hg, but it does not emphasise the discussion on human health outcomes relating to data evaluation, evidence-based data for intervention and disease risk. The paper does discuss the use of blood levels as biomarkers for assessing elevated levels of Se and Hg from dietary exposure.

Reviewer response: To me, the relation between blood concentrations as biomarkers and adverse health outcomes is critical for interpreting the link between dietary exposures to health outcomes, and its absence reduces the review quality. Do the authors have a rationale for excluding this? If data is limited, this would still be valuable to discuss as a needed research direction.

 Response: Except for accidental exposure of Hg from contaminant food and inhalation from pollutant work environment, there has been little information on the link of dietary Hg exposure on adverse health effect as measured by the blood biomarker in humans. The recent article by Kuras et al (2019), described the effect of Hg exposure on healthy volunteers including information on effects of dietary Hg exposure from fish consumption, and the findings of this study have been included in the text (highlighted in blue, lines 243-258). The discussion on Se biomarkers for high expose of dietary Se has also been added in the text (highlighted in blue, lines 206-210).

#2 (Previous 2) Please add stated methods to the manuscript and include what search terms were included for PubMed and ScienceDirect, including the date databases were searched and the total number of retrieved results by database.

Response: The methods of literature search using PubMed and ScienceDirect have been added in the text (highlighted in blue, lines 69-86).